# An Azo-Group-Functionalized Porous Aromatic Framework for Achieving Highly Efficient Capture of Iodine

**DOI:** 10.3390/molecules27196297

**Published:** 2022-09-23

**Authors:** Zhuojun Yan, Yimin Qiao, Jiale Wang, Jialin Xie, Bo Cui, Yu Fu, Jiawei Lu, Yajie Yang, Naishun Bu, Ye Yuan, Lixin Xia

**Affiliations:** 1College of Chemistry, Liaoning University, Shenyang 110036, China; 2School of Environmental Science, Liaoning University, Shenyang 110036, China; 3Key Laboratory of Polyoxometalate and Reticular Material Chemistry of Ministry of Education, Faculty of Chemistry, Northeast Normal University, Changchun 130024, China; 4Yingkou Institute of Technology, Yingkou 115014, China

**Keywords:** azo group, porous aromatic framework, radioactive iodine, vapor capture, suzuki reaction

## Abstract

The strong radioactivity of iodine compounds derived from nuclear power plant wastes has motivated the development of highly efficient adsorbents. Porous aromatic frameworks (PAFs) have attracted much attention due to their low density and diverse structure. In this work, an azo group containing PAF solid, denoted as LNU-58, was prepared through Suzuki polymerization of tris-(4-(4,4,5,5-tetramethyl-1,3,2-dioxaborolan-2-yl)-phenyl)-amine and 3,5-dibromoazobenzene building monomers. Based on the specific polarity properities of the azo groups, the electron-rich aromatic fragments in the hierarchical architecture efficiently capture iodine molecules with an adsorption capacity of 3533.11 mg g^−1^ (353 wt%) for gaseous iodine and 903.6 mg g^−1^ (90 wt%) for dissolved iodine. The iodine uptake per specific surface area up to 8.55 wt% m^−2^ g^−1^ achieves the highest level among all porous adsorbents. This work illustrates the successful preparation of a new type of porous adsorbent that is expected to be applied in the field of practical iodine adsorption.

## 1. Introduction

Nuclear energy is a clean and efficient energy source, which is expected to replace traditional fossil fuels and become a new energy stock group. The Nuclear Energy Agency of the United States predicts an 80% increase in nuclear energy by 2050 with over 715 GW(e)/annum [1]. With the rapid development of the nuclear energy industry, the emission of radioactive substances has attracted significant attention and concern. The discharge of radioactive iodine species is particularly noticeable due to rapid mobility and significant bioaccumulation [2]. For example, ^129^I has an extraordinary half-life (1.57 × 10^7^ years) [3,4], its compounds are usually expelled by gas into the natural environment, then deposited into the food chain, and eventually absorbed by living animals. Once in the body, radioactive iodine substances can affect the normal metabolism of the human body, causing coma, thyroid disease, and even cancer [1,5,6].

To solve this problem, efforts have been made to develop adsorbent materials that can capture radioactive iodine waste in both the gaseous and aqueous phases. For example, various metal–organic frameworks (MOFs) [7,8], conjugated microporous polymers (CMPs) [9,10], and covalent organic frameworks (COFs) [11,12] have been prepared that feature the use of various building blocks as well as tunable pore structure that have demonstrated excellent capabilities for I_2_ capture.

In recent years, porous aromatic frameworks (PAFs) are a new type of porous materials that are constructed from carbon–carbon-bonded aromatic building units and possess rigid frameworks and exceptionally high surface areas [13,14,15]. Various functions can be achieved either from the intrinsic chemistry of their building units or by post-modification [16,17,18,19]. In recent years, PAFs have made significant progress in the fields of energy storage [20,21], catalysis [22,23], adsorption [24,25], etc. In our previous work, we tried to improve the binding capacity and adsorption performance of iodine molecules by changing the pore structure and structural components. For example, we introduced the N atom and π-conjugated structure, and realized a strong affinity and high capacity for iodine molecules [26]. In this work, 3,5-dibromoazobenzene was adopted as the building unit to prepare an azo-containing PAF solid through a one-pot Suzuki coupling reaction, denoted as LNU-58. The introduced azo groups provide polarity properties and also possess electron-rich π-conjugated fragments for interactions with I_2_ molecules. Consequently, the resulting azo-group-functionalized PAF solid shows excellent capture capacity for gaseous and dissolved iodine.

## 2. Results

### 2.1. Structure Characterizations of LNU-58

LNU-58 was prepared using tris-(4-(4,4,5,5-tetramethyl-1,3,2dioxaborolan-2-yl)-phenyl)-amine and 3,5-dibromoazobenzene as raw materials via a Suzuki coupling reaction (Figure 1). Fourier transform infrared (FT-IR) spectroscopy was used to characterize the bonding structure of LNU-58. The B-O and C-Br peaks at 1366 and 497 cm^−1^ disappeared, respectively, which demonstrated the complete crosslinking of the Suzuki polymerization [27]. As for the characteristic band for -N=N- groups, it was found at 1450 cm^−1^ [28] (Figure 2a), verifying structural intact azo groups in the LNU-58 network. ^13^C solid-state NMR (CP/MAS) spectroscopy was applied to investigate the local structure of LNU-58 (Figure 2b). The distinct signals in the range from 110 to 155 ppm were attributed to the aromatic carbon atoms; thereinto, the peak at around 152 ppm was assigned to the carbon atoms attached to the azo groups [27,28,29]. The FT-IR and NMR results both proved the successful synthesis of the LNU-58 network.

Scanning electron microscopy (SEM) and transmission electron microscopy (TEM) were used to study the morphology of LNU-58 (Figure 2c). SEM imagery illustrated that LNU-58 powder was basically composed of a high number of randomly distributed nanospheres. The TEM image showed the amorphous structure of LNU-58 (Figure 2c insert). These observations are consistent with the analysis of powder X-ray diffraction (PXRD) in Figure 2d, demonstrating no long-range ordered structure of the PAF framework.

A thermogravimetric analysis (TGA) was conducted under air atmosphere condition to test the thermal stability of LNU-58 (Figure 3a). It is worth noting that the LNU-58 did not show any significant weight loss until ~300 °C. In addition, LNU-58 PAF could not be dissolved or decomposed in various solvents, including methanol, ethanol, acetone, dichloromethane, chloroform, DMF, tetrahydrofuran, etc., which suggested excellent thermal and solvent stability.

The nitrogen adsorption–desorption isotherm of LNU-58 was recorded at 77 K, which has been defined as a Type IV isotherm (Figure 3b) [30,31]. As the pressure increased, the LNU-58 adsorption isotherm revealed a slight hysteresis, indicating the existence of both micro- and mesopore cavities in the PAF architecture [32]. This hysteresis loop was attributed to the dynamic behavior of azo groups in the LNU-58 framework through an elastic deformation during the course of the gas adsorption [33,34]. Using the Brunauer–Emmett–Teller (BET) calculation model, it was found that the specific surface area of LNU-58 was ~41 m^2^ g^−1^. The pore size distribution (PSD) calculated by non-local density functional theory (NLDFT) was located in the range of 1.8–5.3 nm (Figure 3b insert). The Calc. Model was N_2_ at 77 K on carbon (slit pore, QSDFT equilibrium model), and the relative press range was 0.0000–1.0000. The lower confidence limit was 0.614 nm. These micro-sized apertures were capable of storing iodine guests, and the mesopores facilitated their diffusion by decreasing the mass-transfer resistance [2].

### 2.2. Iodine Adsorption Study

The adsorption performance of LNU-58 on both gaseous and dissolved iodine was further explored by calculating the weight of adsorbent solids before and after iodine adsorption by gravimetric analysis. The adsorption uptake of LNU-58 was close to linear at the initial stage (0–10 h) of adsorption and it achieved a saturation state with a sufficient contact time (50 h). As shown in Figure 4a inset, LNU-58 powder changed from yellow to dark brown and the maximum iodine uptake capacity of LNU-58 was up to 3533.11 mg g^−^^1^ (Figure 4a). Additionally, the LNU-58 sample was capable of trapping iodine from an aqueous solution with a high capacity of 903.6 mg g^−1^ in 12 h (Figure 4b).

Notably, the adsorption of iodine on the PAF powder was reversible. The captured iodine could be easily eluted from the iodine-exhaust sample by placing the LNU-58 powder in ethanol at room temperature for 48 h. As depicted in Figure 4c, the solution changed from colorless to deep brown, which demonstrated the release of iodine into ethanol. Calculated by the BET surface area (~41 m^2^ g^−^^1^), the iodine uptake per specific surface area is ~8.55 wt% m^−^^2^ g^−1^. The iodine adsorption capacity per specific surface area of LNU-58 exceeds that of most other porous adsorbents (Figure 4d) [35,36,37,38,39,40,41,42,43,44,45,46].

To verify the mechanism of the iodine adsorption, PXRD, FT-IR, and Raman spectra were performed to detect the interactions between the LNU-58 and I_2_. As shown in Figure 5a, the adsorption process fitted by the pseudo-second-order adsorption kinetic model indicated the chemical adsorption of PAF adsorbent for I_2_ molecules [2,46]. As shown in the PXRD patterns, there is no characteristic diffraction peaks for I_2_ crystalline, which demonstrates the uniform distribution of iodine molecules in the LNU-58 framework (Figure 5b) [26,46].

The characteristic FT-IR bands of C=C and C-H (phenyl ring) in the LNU-58@I_2_ changed from 1509 and 830 cm^−1^ to 1506 and 825 cm^−1^, respectively, demonstrating the strong interactions of aromatic fragments with I_2_ molecules (Figure 5c). Meanwhile, the N=N band shifted from 1450 (pristine LNU-58) to 1448 cm^−1^ after iodine was loaded [26,27,46], which indicated that iodine formed a charge transfer complex with the LNU-58, thereby, promoting adsorption efficiency for iodine molecules. As shown in Raman spectroscopy (Figure 5d), the characteristic peaks at around 110, 135, and 169 cm^−1^ belonged to the perturbed di-iodine molecules and asymmetric I_3_^−^ ions for polyiodides [I_3_^−^ I_2_]; meanwhile, the peaks at 147 and 169 cm^−1^ were ascribed to the perturbed di-iodine molecules of [I^−^·(I_2_)_2_] [47,48]. Apart from that, the emerging peak at 169 cm^−1^ assigned to the I_5_^−^ compound indicated the charge transfer between the guest (iodine molecules) and host network (LNU-58) [49].

## 3. Experimental

### 3.1. Materials

3,5-Dibromoazobenzene was synthesized according to the literature [50]. Tris-(4-(4,4,5,5-tetramethyl-1,3,2-dioxaborolan-2-yl)-phenyl)-amine was purchased from Energy Chemical. Tetrakis(triphenylphosphine)palladium and potassium carbonate were purchased from TCI. Other chemical reagents and solvents were obtained from commercial suppliers and used directly.

### 3.2. Characterization

Fourier transfer infrared (FT-IR) spectrum was performed on a Shimadzu-Prestige21 spectrometer in the range of 400–4000 cm^−1^. Solid-state ^13^C NMR spectra were obtained using a Bruker Avance III model 400 MHz NMR spectrometer at a MAS rate of 5 kHz. Scanning electron microscope (SEM) imagery was obtained using a SU8010 model scanning electron microscope with an accelerating voltage of 5 kV. The PAF samples were prepared by dispersing the LNU-58 powder onto a silicon wafer. The transmission electron microscopy (TEM) imagery was recorded using a JEM-2100 with an accelerating voltage of 200 kV. The textural properties were obtained by carrying out N_2_ adsorption/desorption experiments using a Micromeritics ASAP 2460 instrument at 77 K; the samples were dried for 10 h in a vacuum environment at 90 °C before the test. The thermogravimetric analysis (TGA) was performed using a TGA/DSC 2 thermogravimetric analyzer under air atmosphere conditions with a heating rate of 10 °C min^−1^ and a temperature range of 20–800 °C. The powder X-ray diffraction (PXRD) pattern was collected in the 2θ range of 5–60 with a scan speed of 5 ° min^−1^ using a Bruker D8 ADVANCE diffraction.

### 3.3. Synthesis of LNU-58

Tris-(4-(4,4,5,5-tetramethyl-1,3,2dioxaborolan-2-yl)-phenyl)-amine (243.67 mg, 0.391 mmol) and 3,5-dibromoazobenzene (200 mg, 0.588 mmol) were dissolved in N,N′-dimethylformamide (60 mL) solution, and then added into a two-neck flask equipped with a condenser. After three freeze-pump-thaw cycles, K_2_CO_3_ aqueous solution (2 M, 5 mL) and tetrakis(triphenylphosphine)palladium (40 mg 0.035 mmol) were quickly added to the system and degassed by three freeze-pump-thaw cycles; the mixture was stirred at 130 °C for 48 h. After cooling to room temperature, the residue was filtered and washed with tetrahydrofuran (THF), water (H_2_O), and dichloromethane (DCM) to remove all unreacted monomers and catalysts. Finally, further purification of the PAF product was carried out via Soxhlet extraction using tetrahydrofuran (THF) and dichloromethane (DCM), in turns, for 48 h. The product was dried under vacuum at 90 °C to obtain the LNU-58 powder (~86.0% yield).

### 3.4. Iodine Adsorption and Release Experiment

#### 3.4.1. Iodine Vapor Uptake Experiments

LNU-58 was degassed at 90 °C for 12 h under a vacuum before the experiment. The iodine vapor uptakes of PAF solid were tested using the gravimetric analysis. LNU-58 (30 mg) and excess iodine were loaded into an open weighing bottle, which was then placed in a closed system at 75 °C under ambient pressure. The iodine sorption capacity was calculated by the following Formula (1):(1)$$E_{u}=\frac{m_{2}-m_{1}}{m_{1}}\times 100\%$$
where *m*_1_ and *m*_2_ are the masses of PAF powder before and after iodine uptake, respectively.

#### 3.4.2. Dissoved Iodine Uptake Experiments

Before the experiment, LNU-58 was degassed at 90 °C for 12 h under a vacuum. The dissolved iodine uptakes of PAF solid were tested using the gravimetric analysis. LNU-58 (50 mg) was immersed in two glass vials containing KI aqueous solution (600 mg KI in 3 mL of H_2_O) and KI_3_ aqueous solution (300 mg I_2_ and 600 mg KI in 3 mL of H_2_O) at room temperature for 48 h, respectively. At each time interval, the used material was vacuum filtered through an organic nylon filter membrane, then washed with water under air atmosphere conditions, dried with qualitative filter paper from the aqueous solution adhering to the material, and weighed.

#### 3.4.3. Iodine Desorption

Ethanol was used as the extraction solvent to evaluate the reversibility of the iodine adsorbed in the PAF material. Five milligrams of iodine-loaded polymer was poured into five milliliters of ethanol; the release process of iodine was photographed at selected time intervals.

## 4. Conclusions

In summary, an azo-group-functionalized porous aromatic framework with the specific polarity properties of azo groups and electron-rich aromatic fragments was prepared through one-step Suzuki polymerization. Based on this specified structure, the resulting solid realized ultra-high iodine capture up to 3533.11 mg g^−1^ (353 wt%). Our work firmly demonstrates the importance of the azo-rich backbone in iodine capture which provides a promising candidate for radioactive iodine capture and sequestration to address environmental issues.

## Figures and Tables

**Figure 1 molecules-27-06297-f001:**
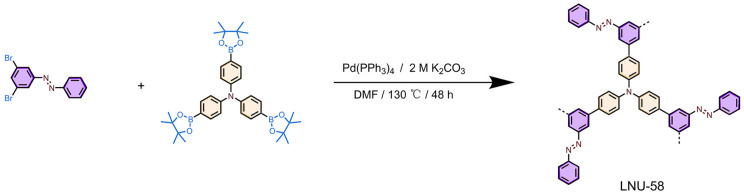
Synthesis and structure of LNU-58.

**Figure 2 molecules-27-06297-f002:**
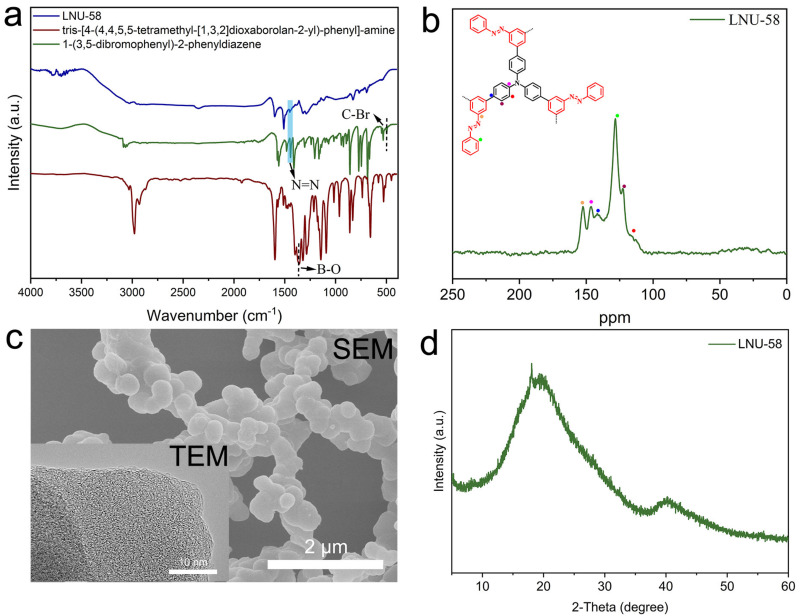
(**a**) FT-IR spectra of LNU-58 and related raw materials; (**b**) solid-state ^13^C NMR spectrum of LNU-58; (**c**) SEM image and TEM image (insert) of LNU-58; (**d**) PXRD pattern of LNU-58.

**Figure 3 molecules-27-06297-f003:**
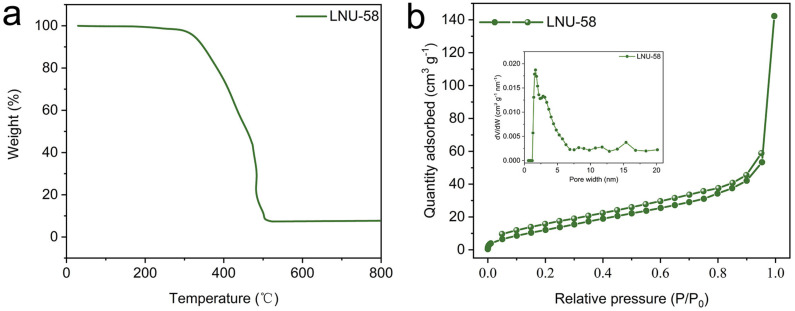
(**a**) TGA plot for LNU-58 under air atmosphere condition; (**b**) N_2_ adsorption–desorption isotherm of LNU-58 at 77 K, pore size distribution of LNU-58 (insert).

**Figure 4 molecules-27-06297-f004:**
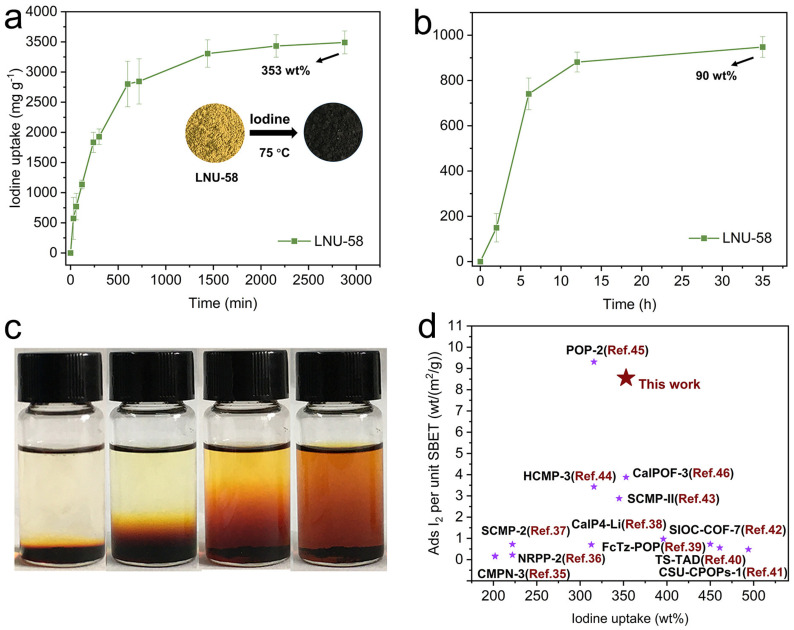
(**a**) Gravimetric I_2_ uptake of LNU-58 as a function of time at 75 °C. Photograph inserts show the color change of LNU-58 before and after iodine sorption; (**b**) iodine uptake of LNU-58 as a function of time from iodine aqueous solution; (**c**) photographs indicate gradual changes in iodine desorption processes of LNU-58; (**d**) comparison of iodine uptake capacities with other bench solids.

**Figure 5 molecules-27-06297-f005:**
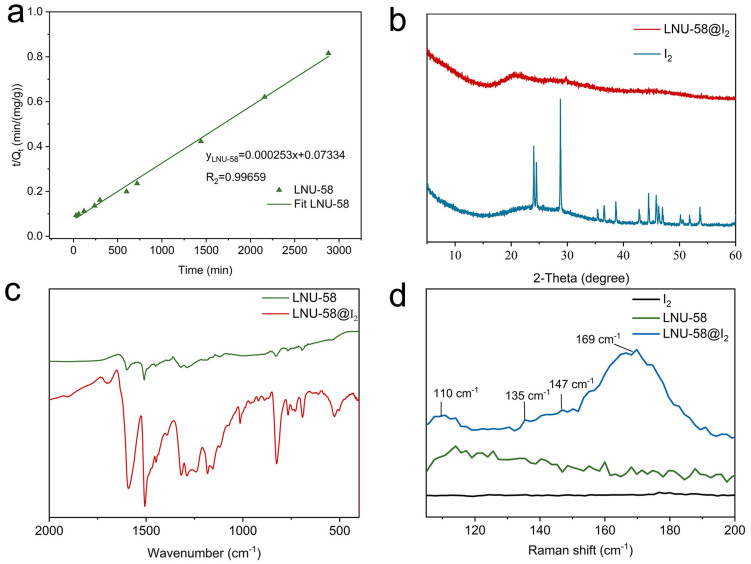
(**a**) Pseudo-second-order model plot for the iodine vapor uptake into LNU-58; (**b**) PXRD patterns of I_2_ and LNU-58@I_2_; (**c**) FT-IR spectra of LNU-58 and LNU-58@I_2_; (**d**) Raman spectra of I_2_, LNU-58, and LNU-58@I_2_.

## Data Availability

All data related to this study are presented in this publication.

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
