# Peer review of "An Azo-Group-Functionalized Porous Aromatic Framework for Achieving Highly Efficient Capture of Iodine"

_molecules, 2022, doi:10.3390/molecules27196297_

Round 1

Reviewer 1 Report

This is an interesting paper describing the synthesis and characterisation of an azo-functionalised PAF suitable for the encapsulation of iodine. THe new compound has been characterised vis pectroscopic techniques as well as SEM and TEM and its absorption capability has been investigated. It is a well-written paper reporteing on an interesting piece of research. Two minor comments: 1. The introduction is short. More information can be added especially on the properties, applications, ets of PAFs. 2. Have the authors performed thermodynamic studies for the iodine adsorption? That would be useful in order to find the maximum adsorption capacity depending on the PAF/I ratio.

Author Response

Reviewer 1

Question 1: The introduction is short. More information can be added especially on the properties, applications, ets of PAFs.

Response 1: Thanks for your suggestion, which helps us a lot to improve the quality of the manuscript. According to the suggestion, we have made change to the manuscript.

“Porous aromatic frameworks (PAFs), a new type of porous material in recent years, are constructed from carbon-carbon-bonded aromatic building units which possess a rigid framework and an exceptionally high surface area [13-15]. Various functions can be achieved either from the intrinsic chemistry of their building units or by post-modification [16-19]. In recent years, PAFs have made great progress in the fields of energy storage [20,21], catalysis [22,23] and adsorption [24,25], etc” (Please see Lines 49-54, page 2 in the revised manuscript)

Question 2: Have the authors performed thermodynamic studies for the iodine adsorption? That would be useful in order to find the maximum adsorption capacity depending on the PAF/I ratio

Response 2: Thanks for your question. Your suggestions are very helpful for improving the quality of our article. However, our laboratory was closed due to the epidemic. The specific opening time has not yet been determined, and we will conduct thermodynamic studies for the iodine adsorption in our follow-up work.

Reviewer 2 Report

In this contribution by Yan et al., the porous aromatic framework, PAF, LNU-58 has been synthesised and characterised well. The authors report iodine capture properties of LNU-58, which makes the compound interesting to look at. Overall, I will be glad to reconsider this for Molecules, subject to the following major revisions:

1)    Page 2, line 46: the authors write “nitrate-impregnated silicic acid and silver-exchanged molecular sieve, etc.).” to exemplify page 1, line 45 “variety of commercially available solid sorbents”. This is not the full picture of the state-of-the-art in I2 capture, because in the grand scheme of things, to replace the cost- and energy-intensive state-of-the-art, alternative, energy-efficient solid adsorbents, such as, metal-organic frameworks (MOFs), conjugated microporous polymers (CMPs), and covalent organic frameworks (COFs) for I2 capture are known from a long time. For context, MOFs are known in this area for ca. 20 years now, earliest report being in Angewandte Chemie, 2003, DOI: 10.1002/anie.200250633, and follow-up reports include, CrystEngComm, 2013, DOI: 10.1039/C3CE40795K. CMPs and COFs are not far behind as well, Chemcomm, 2014, DOI: 10.1039/C4CC01783H; Chem. Commun., 2017, DOI: 10.1039/C7CC01045A. I advise the authors to expand upon the introduction section (thanks to these citations), covering these bases first over 1-2 manuscript sentences, before transitioning into the current class of sorbents, PAFs.

2)    It is interesting that LNU-58 is nonporous to N2 at 77 K. Albeit this non-porosity revealed in Figure 3b, Figure 3b inset reveals biporosity, micro- and mesopores together. I strongly advise the authors to provide details on how they determined the pore size distribution plot, because the fit model does not seem to be in perfect agreement with the nonporous Type IV N2 isotherm. If it’s indeed biporous in nature, this is where the current results with a PAF prototype needs to be contextualised in comparison to previous reports of biporous MOFs/COFs/CMPs for I2 capture, see CrystEngComm, 2013, DOI: 10.1039/C3CE40795K as an example.

3)    Figure 4a and 4b need triplicate measurements at all the data points, and each of these should include error bars. This is a must to ensure data reproducibility.

4)    Figure 4d, font sizes within the plot needs to be bigger for improved clarity.

5)    Page 6, lines 200-201, the authors write “At each time interval, the exhausted material was filtered and then washed, and the aqueous solution attached to the material was blotted dry and weighed”. This is far from satisfactory, in terms of the experimental details. Please specify which filter paper, what was the washing solvent (washed under vacuum / under air, or not / neither), how dried, air dried / or otherwise etc.). How did the authors ensure that the Iodine wasn’t physically adhering on top of the LNU-58 solid powder? That’s where the step of washing is most critical, and needs to be emphasised.

Author Response

Review 2

Question 1: Page 2, line 46: the authors write “nitrate-impregnated silicic acid and silver-exchanged molecular sieve, etc.).” to exemplify page 1, line 45 “variety of commercially available solid sorbents”. This is not the full picture of the state-of-the-art in I2 capture, because in the grand scheme of things, to replace the cost- and energy-intensive state-of-the-art, alternative, energy-efficient solid adsorbents, such as, metal-organic frameworks (MOFs), conjugated microporous polymers (CMPs), and covalent organic frameworks (COFs) for I2 capture are known from a long time. For context, MOFs are known in this area for ca. 20 years now, earliest report being in Angewandte Chemie, 2003, DOI: 10.1002/anie.200250633, and follow-up reports include, CrystEngComm, 2013, DOI: 10.1039/C3CE40795K. CMPs and COFs are not far behind as well, Chemcomm, 2014, DOI: 10.1039/C4CC01783H; Chem. Commun., 2017, DOI: 10.1039/C7CC01045A. I advise the authors to expand upon the introduction section (thanks to these citations), covering these bases first over 1-2 manuscript sentences, before transitioning into the current class of sorbents, PAFs.

Response 1: Thanks for your suggestion, which helps us a lot to improve the quality of the manuscript. According to the suggestion, we have made change to the manuscript.

“For example, a various of metal-organic frameworks (MOFs) [7,8], conjugated microporous polymers (CMPs) [9,10] and covalent organic frameworks (COFs) [11,12] have been prepared featuring by various building blocks as well as tunable pore structure, which illustrate excellent capability for I2 capture.” (Please see Lines 44-48, page 2 in the revised manuscript)

Question 2: It is interesting that LNU-58 is nonporous to N2 at 77 K. Albeit this non-porosity revealed in Figure 3b, Figure 3b inset reveals biporosity, micro- and mesopores together. I strongly advise the authors to provide details on how they determined the pore size distribution plot, because the fit model does not seem to be in perfect agreement with the nonporous Type IV N2 isotherm. If it’s indeed biporous in nature, this is where the current results with a PAF prototype needs to be contextualized in comparison to previous reports of biporous MOFs/COFs/CMPs for I2 capture, see CrystEngComm, 2013, DOI: 10.1039/C3CE40795K as an example.

Response 2: Thanks for your question. The Calc. Model is N2 at 77 K on carbon (slit pore, QSDFT equilibrium model) and the rel. press range is 0.0000 - 1.0000. The lower confidence limit is 0.614 nm. All these details were added to the revised manuscript. (Please see Lines 106-108, page 3 in the revised manuscript)

Similar results of porous solids with biporosity are contextualized in comparison to previous reports of biporous MOFs/COFs/CMPs for I2 capture, including: ACS Appl. Polym. Mater. 2020, 2, 11, 5121-5128; Chem. Sci. 2011, 2, 1777-1781; Micropor. Mesopor. Mat. 2016, 231, 92-99; Polymer 2014, 55, 321-325.

Question 3: Figure 4a and 4b need triplicate measurements at all the data points, and each of these should include error bars. This is a must to ensure data reproducibility.

Response 3: Thanks for your suggestion. According to your comment, we carefully conducted the analysis for the I2 capture and included error bars in Figure 4a-b.

Question 4: Figure 4d, font sizes within the plot needs to be bigger for improved clarity.

Response 4: Thanks for your question. We have increased the font size in Figure 4d.

Question 5: Page 6, lines 200-201, the authors write “At each time interval, the exhausted material was filtered and then washed, and the aqueous solution attached to the material was blotted dry and weighed”. This is far from satisfactory, in terms of the experimental details. Please specify which filter paper, what was the washing solvent (washed under vacuum / under air, or not / neither), how dried, air dried / or otherwise etc.). How did the authors ensure that the Iodine wasn’t physically adhering on top of the LNU-58 solid powder? That’s where the step of washing is most critical, and needs to be emphasised.

Response 5: Thanks for your suggestion. The operation steps mainly refer to previous reports (J. Mater. Sci. 2020, 55, 1854-1864; Mater. Chem. Phys. 2020, 239, 122328). According to your advice, we have made a change to the manuscript.

“At each time interval, the used material was vacuum filtered through an organic nylon filter membrane, then washed with water under air conditions, dried with qualitative filter paper from the aqueous solution adhering to the material, then weighed.” (Please see Line 205-208, page 6 in the revised manuscript)

Round 2

Reviewer 2 Report

I am happy with all revisions made by the authors, will be glad to see it accepted in Molecules.